# Improving Retention in Mental Health and Psychosocial Support Interventions: An Analysis of Completion Rates across a Multi-Site Trial with Refugee, Migrant, and Host Communities in Latin America

**DOI:** 10.3390/ijerph21040397

**Published:** 2024-03-25

**Authors:** Isabella Fernández Capriles, Andrea Armijos, Alejandra Angulo, Matthew Schojan, Milton L. Wainberg, Annie G. Bonz, Wietse A. Tol, M. Claire Greene

**Affiliations:** 1Heilbrunn Department of Population and Family Health, Columbia University Mailman School of Public Health, New York, NY 10032, USA; mg4069@cumc.columbia.edu; 2HIAS Ecuador, Quito 170143, Ecuador; andrea.armijos@hias.org; 3HIAS Panamá, Panamá City, Panamá; alejandra.angulo@hias.org; 4HIAS, Silver Spring, MD 20910, USA; matthew.schojan@hias.org (M.S.); annie.bonz@hias.org (A.G.B.); 5Department of Psychiatry, New York State Psychiatric Institute, Columbia University, New York, NY 10032, USA; milton.wainberg@nyspi.columbia.edu; 6Global Health Section, Department of Public Health, University of Copenhagen, 1172 Copenhagen, Denmark; wietse.tol@sund.ku.dk; 7Athena Research Institute, Vrije Universiteit Amsterdam, 1081 HV Amsterdam, The Netherlands

**Keywords:** MHPSS interventions, retention, completion rate, refugee and migrant communities, feasibility

## Abstract

Research on mental health and psychosocial support (MHPSS) interventions within refugee and migrant communities has increasingly focused on evaluating implementation, including identifying strategies to promote retention in services. This study examines the relationship between participant characteristics, study setting, and reasons for intervention noncompletion using data from the Entre Nosotras feasibility trial, a community-based MHPSS intervention targeting refugee, migrant, and host community women in Ecuador and Panama that aimed to promote psychosocial wellbeing. Among 225 enrolled women, approximately half completed the intervention, with varying completion rates and reasons for nonattendance across study sites. Participants who were older, had migrated for family reasons, had spent more time in the study community, and were living in Panamá (vs. Ecuador) were more likely to complete the intervention. The findings suggest the need to adapt MHPSS interventions to consider the duration of access to the target population and explore different delivery modalities including the role of technology and cellular devices as reliable or unreliable source for engaging with participants. Engaging younger, newly arrived women is crucial, as they showed lower completion rates. Strategies such as consulting scheduling preferences, providing on-site childcare, and integrating MHPSS interventions with other programs could enhance intervention attendance.

## 1. Introduction

According to the International Organization for Migration, the global population of international migrants reached nearly 281 million people in 2020, accounting for approximately 3.6% of the world’s population [1]. The number of migrants residing in the Latin America and Caribbean regions has experienced a significant surge, increasing from an estimated 8.4 million in 2015 to 12.8 million in 2019—a notable growth of over 50% [2]. One major factor contributing to this trend is the ongoing humanitarian crisis, political unrest, and socio-economic instability in Venezuela, which has forced a substantial number of Venezuelans to flee their home country [2,3]. Today, Venezuelans form one of the largest refugee and migrant communities in in the world [3].

The mental health and psychosocial consequences of displacement have been extensively documented [4,5]. Refugees and migrants encounter a range of stressors, including traumatic experiences, poverty, and the breakdown of social support systems [5,6,7]. These factors can have a detrimental impact on their psychosocial well-being and increase the risk of developing mental disorders [7]. Prior to the COVID-19 pandemic, 22% of individuals impacted by displacement reported depression, anxiety, post-traumatic stress disorder (PTSD), bipolar disorder, or schizophrenia—more than triple the rate in the general global population (7%) [8,9]. Similar findings have also been reported among refugee and migrant Venezuelans [10,11]. Consequently, there is an urgent need to adapt and evaluate evidence-based mental health and psychosocial support (MHPSS) interventions to serve populations affected by humanitarian crises [12,13].

Research on MHPSS interventions in humanitarian settings has significantly expanded in recent years [14,15,16]. There has been a notable shift in consensus-based research priorities from examining effectiveness to asking implementation-related questions [14,16]. Rather than solely documenting the impact of structured interventions, the field is now inclined towards understanding how interventions can adapt to the diversity of humanitarian contexts [14,15]. Answering this question necessitates an exploration of how proven MHPSS interventions perform differently depending on the setting. Feasibility studies emerge as crucial tools, frequently employed to test interventions and adapt measures and procedures before undertaking larger-scale trials [17]. Within these trials, tracking data such as intervention initiation, completion, and reasons for non-attendance to intervention sessions becomes essential. This data can be used to evaluate and address challenges related to participant engagement and retention [17,18]. However, given the complex nature in which these interventions are delivered, clear reporting of this information is frequently omitted, making it difficult to understand the feasibility of these types of interventions across diverse contexts [19,20]. Further exploring the factors influencing intervention initiation, attendance, and completion across diverse settings can yield valuable insights into optimizing the implementation and delivery of MHPSS interventions to the unique needs of heterogenous contexts. 

### 1.1. The Entre Nosotras Feasibility Trial in Ecuador and Panamá

To address the mental health and psychosocial needs of Venezuelan women, the Program on Forced Migration and Health at the Heilbrunn Department of Population and Family Health, in collaboration with HIAS (an international non-governmental humanitarian organization), developed and piloted a community based MHPSS intervention [21]. This intervention, called “Entre Nosotras” (meaning “among/between us” in Spanish), consisted of a five-session program aimed at addressing the psychological and social aspects of well-being among refugee and migrant women in Ecuador and Panama [21]. In Ecuador, host community women were also included since their integration emerged as a community and organizational priority during the formative research. The intervention was carefully crafted through a formative qualitative research process and extensive community consultation in both countries, ensuring alignment between the needs expressed by community members and evidence-based intervention principles and strategies to address those needs [21]. The resulting intervention was tailored to the target population and tackled a range of social challenges, including interpersonal violence, xenophobia, and social isolation, as well as psychological problems like emotional distress and sadness. A manual was developed to guide the implementation of each Entre Nosotras session, providing a clear outline of the core elements and activities [21]. This approach aimed to ensure fidelity to the intervention model across different sites while allowing for adaptability to diverse contexts [21].

Between September 2021 and March 2022, a feasibility trial was conducted across multiple study settings with 225 women. The primary objective of the trial was to assess the appropriateness, acceptability, safety, and feasibility of conducting a fully powered cluster randomized trial for the Entre Nosotras intervention. A comprehensive protocol outlining the details of the feasibility trial was published and registered online at: https://clinicaltrials.gov/study/NCT05130944 (accessed on 17 June 2023). The primary outcomes of the trial are available elsewhere. Throughout the trial, data on session attendance and reasons for missing any session were systematically collected. 

The Entre Nosotras feasibility trial spanned eleven communities located within three distinct sites: Guayaquil [*n* = 72] and Tulcán [*n* = 71] in Ecuador, and Panamá City/Panamá West [*n* = 82] in Panamá. Guayaquil is a large, urban, coastal city that attracts numerous migrants from Colombia and Venezuela and has a large informal labor market [22]. Tulcán, situated in the highlands of Ecuador on the border with Colombia, is a rural city that frequently serves as a temporary transit point for migrants and refugees. Panamá City is the capital city of Panamá and features a growing labor market [23]. It is the preferred destination for migrants primarily from Central and South America; many migrants settle in the peri-urban areas surrounding Panamá West. The heterogeneous nature of the study settings presents a valuable opportunity to explore variations in intervention completion and reasons for not attending intervention sessions.

### 1.2. Objective

The goal of this study was to use tracking data on attendance and reasons for non-attendance to intervention sessions from the Entre Nosotras feasibility trial to examine correlations between intervention initiation and completion and describe the reasons for and variations in missed sessions by study setting.

## 2. Materials and Methods

This study involved a secondary analysis of data from the cluster randomized feasibility trial of the Entre Nosotras intervention [21].

### 2.1. Participants and Procedures

The sample included the 225 women enrolled in the Entre Nosotras feasibility trial. Participants were eligible for the parent study if they were 18+ years of age, identified as a woman, were currently residing in the study community, spoke and understood Spanish, and reported up to moderate psychological distress. Psychological distress was assessed using the Kessler-6 scale, a brief screening tool designed to measure non-specific psychological distress [24]. Participants reporting up to moderate levels of distress, indicated by a Kessler-6 score of less than 13, were deemed eligible for inclusion in the study. Individuals who reported severe psychological distress (Kessler-6 score ≥ 13), and thus may have required more specialized care, were referred to HIAS for further evaluation and services. Participants were recruited through referral from HIAS staff, community outreach workers, and community leaders. Study research assistants screened all participants for eligibility prior to enrollment. Baseline assessments were completed within 1 week of enrollment and within 2 weeks of completing the baseline, participants attended the first Entre Nosotras Session. 

### 2.2. Measures

Using session attendance data, participants were classified into one of three groups: those who completed the intervention (“Completed intervention”, 4–5 sessions), those who did not complete the intervention (“Partial attendance”, 1–3 sessions), and those who provided baseline data but did not attend any intervention sessions (“Never started”, 0 sessions). These groups will be compared based on baseline characteristics, including age, study site, nationality, education, employment, reasons for migration, and length of time in the community. To address non-attendance, the Entre Nosotras staff proactively reached out to participants who missed any session and collected information on the reasons for their absence. This was carried out continuously, as part of routine program implementation during the study period. The responses to this open- ended question were recoded into categories within the reason for missing variable. These categories were utilized to determine the main reasons for non-completion of the intervention among participants who partially attended or never initiated it.

### 2.3. Analytical Methods

Using data from the Entre Nosotras feasibility Trial we analyzed and estimated the proportions of participants who completed the intervention (4–5 sessions), partially engaged in the intervention (1–3 sessions), or did not attend any intervention sessions (0 sessions). To compare the baseline characteristics among the three study groups, several bivariate analyses were conducted using appropriate statistical tests. Age was compared across intervention completion groups using ANOVA, while Fisher’s exact test or the Chi-square test were employed for categorical variables such as study site, participant’s nationality, education, employment, reasons for migration, and time in the community. P-values were reported to indicate significant differences (*p* < 0.05) between the groups. To determine the reasons for missing each session, a content analysis was performed on the notes taken by staff members in routine study attendance logs. This analysis involved examining the notes for recurring words or concepts, which were subsequently recoded into the same category. The lead author carried out the preliminary coding and labeled the categories based on a common representative theme. This was reviewed by the senior author and then consulted with the research assistants in the different study localities. The main reason for not completing the intervention is a constructed variable, which was identified when participants missed two or more sessions for the same reason and was compared across the Partial Attendance and Never Started groups.

## 3. Results

Table 1 presents the characteristics of the study population enrolled in the Entre Nosotras feasibility trial, categorized by their completion status. A total of 225 women were included in the study, with nearly half (49.8%) completing the intervention, more than a quarter (28.0%) who completed 1–3 sessions, and nearly a quarter (22.2%) never starting the intervention.

The mean age of the participants was 36.0 years (SD = 11.7) with a significant difference observed among the completion status groups (*p* = 0.001). Participants who completed the intervention were significantly older (38.9 years, SD =12.7) than those who partially attended (32.9 years, SD = 9.8, *p* = 0.002, see Appendix A, Table A1), and those who never started the intervention (33.6 years, SD = 9.9, *p* = 0.014, see Appendix A, Table A1). However, no significant difference was observed between those who partially attended, and those who never started the intervention. Study site varied significantly among the completion status groups (*p* = 0.030), with a higher proportion of participants completing the intervention in Panama (43.8%) compared to Guayaquil (25.0%) and Tulcán (31.2%). Tulcán showed a lower proportion of participants with partial attendance (25.4%) compared to Panama (36.5%) and Guayaquil (38.1%), and Panama had the lowest proportion of participants who never started the intervention (20%) compared to Guayaquil (40%) and Tulcán (40%).

Overall, most of participants were Venezuelan (65.9%) followed by Colombian (14.8%), and Ecuadoran (12.6%). More than half of the participants had completed high school (52.0%), and nearly a quarter held a university degree (24.7%). The majority of the participants were unemployed (53.8%). There were no significant differences in the distribution of nationality, education, and employment between completion status groups.

A significant difference in the reasons for migration was observed between completion status groups (*p* = 0.005). The most common reason for migrating was economic troubles (42.3%); however, a higher proportion of participants who completed the intervention migrated due to family reasons (37.0%) compared to those who partially attended (19.0%) or never started the intervention (30.2%).

There was a marginal significant difference in the time living in the community among completion status groups (*p* = 0.052). More than three fourths of participants had been in the community for over 1 year (76.4%). However, a higher proportion of participants who completed the intervention had been in the community for more the 3 years (45.5%) compared to those who partially attended (27.0%) or never started the intervention (28.0%).

The reported reasons for missing sessions were coded into six categories: work or school, family responsibilities, medical incapacity, logistical issues, other personal causes, and unreachable. Table 2 presents a description of each of these categories.

Table 3 shows the reasons for missing any Entre Nosotras intervention session by study site. From the 1125 sessions, there was a total of 484 sessions missed by study participants. The most frequent recorded reason was being unreachable (27.8%), meaning participants could not be contacted to assess the cause for not attending a specific session. This was followed by work or school (24.5%) and other personal causes (18.7%). Notable variations in the distribution of these proportions were observed across the three different study sites. In Tulcán, a higher proportion of participants classified as unreachable were reported (39.7%) as compared to Guayaquil (29.1%) and Panamá (12.6%). Conversely, a higher proportion of participants who reported work or school as their cause for nonattendance were observed in Guayaquil (31.6%) and Panamá (29.6%) compared to Tulcán (12.8%). Other personal causes were also more common in Panamá (28.9%), as compared to Tulcán (15.4%) and Guayaquil (13.3%).

The main reason for non-completion of the intervention, defined as having two or more missed sessions categorized as being for the same reason, are presented in Table 4. Participants having different reasons for all their missing sessions, were considered as having a combination of reasons. Across all participants who partially attended or who never started the intervention (*n* = 113), the main reasons for intervention non-completion were equally distributed between combination of reasons (23.0%), unreachable (23.0%), and work or school (23.0%). However, participants who never started the intervention were more likely to be classified as being unreachable (42.0%) compared to those who were in the partial attendance group (7.9%). A higher proportion of participants who partially attended were classified as having a combination of reasons for missing sessions (34.9%) as compared to the never started group (8.0%). Similar proportions of participants whose main reason was work or school were observed in the partial attendance group (20.6%) and the never started group (26.0%).

## 4. Discussion

Reporting on the retention of participants in MHPSS intervention in humanitarian settings is often omitted or limited [19,20]. However, as the field grows into finding better ways for intervention implementation and delivery, dedicated efforts are needed to understand indicators of implementation such as completion rates. This analysis uniquely explores how participant characteristics, study setting and reasons for nonattendance to intervention sessions relate to completion status.

Our descriptive results suggest that participants who completed the Entre Nosotras feasibility trial are different from those who did not. Those who successfully completed the intervention tended to be older, were more commonly situated in Panamá, had diverse reasons for migrating, and had spent a longer time in the local community. These findings suggest that age, community connectedness, and motivations may influence intervention completion. Consequently, we believe that conducting participatory research with younger, newly arrived women may reveal alternative strategies for enhancing retention and engagement within this subgroup.

The content analysis categorizes and ranks the reasons for missing any session and exposes that these vary across study settings. Being “unreachable” was more frequently recorded in Tulcán, and scheduling conflicts related to work, school, or other personal causes were more frequent in Panamá. The differences in the completion statuses and reasons for nonattendance across the diverse study settings highlight the importance of context when adapting MHPSS interventions. The Entre Nosotras intervention aimed to maintain consistency in implementation across sites while allowing enough flexibility to be adapted to specific populations and contexts. Although this was achieved by using a community participatory approach and having a manual that detailed the core components that were needed to maintain intervention fidelity, we still observed significant variation in intervention completion across sites.

Panama City is a large city where migrants often settle, therefore access to and follow up of participants may be easier. On the other hand, Tulcán is a rural border city that migrants commonly use as a temporary place while they are in transit; for which access to and follow-up of participants for longer periods of time can become challenging. The contrast in how destinations are used by migrants can explain why participants in Tulcán accounted for most of the reasons for missing classified as unreachable, and why Panama had the highest completion rate. This finding supports previous research that suggests difficulties in reaching migrants “in transit” as an important barrier to intervention implementation [25,26]. Although the needs among migrant communities may be similar, reaching a population that is in transit or to which the period of contact is going to be limited will require novel approaches that explore evidence-based interventions that are less time consuming or that can be provided along the way.

Moreover, the main reason for intervention non-completion varied for those who partially attended as compared to those who never started the intervention. Participants who never started the intervention were more frequently unreachable, while participants who partially attended were more frequently classified as having a combination of reasons including work, school, and other personal causes. This requires the field to think of these two groups differently in terms of strategies to improve retention rates. Participants who never started the intervention and whose main reason was being unreachable may be in transit or have other barriers that prevented them from engaging in the intervention. However, other reasons for participants enrolling in the intervention but never attending any session and being classified as unreachable must be considered. Access to certain populations can become difficult in contexts in which women share their phone or own a phone that is controlled by male family members [27]. Also, relying on technology to maintain contact can be difficult because of unstable internet access or network coverage, lack of devices, selling of phones to meet other needs, or low technological literacy [27]. Therefore, we propose including questions in MHPSS intervention recruitment processes that address accessibility to cellular devices, stable internet networks, and phone ownership. Gathering this information will provide a better understanding of the needs of the “unreachable” population and what must be done to maintain a line of communication with them. It is possible that this will not only help with the follow-up of participants but also provide insights to explore if and how technology can be used to expand the reach of MHPSS interventions.

Regarding participants who partially attended the intervention, their main reasons for non-completion varied. Participants’ inability to attend due to work, school, errands, illness, medical appointments, or caregiving responsibilities highlights the importance of offering sessions outside of work hours and exploring alternative options that would make it easier for participants to attend. To ensure participant engagement, it is crucial to consult with them regarding scheduling preferences before the intervention begins. Additionally, efforts should be made to provide on-site childcare services in communities where family responsibilities are identified as a barrier to attendance. This would help to overcome barriers related to childcare responsibilities and facilitate greater participation. Furthermore, this finding also highlights the limited time and competing priorities of migrant and host community women that can impede their participation in this type of intervention. As a result, integrating MHPSS interventions with other support services can enhance overall attendance, contribute to the sustainability of programs, and provide a more efficient and coordinated support system for migrants. 

The findings of this study offer valuable insights that extend beyond the scope of Entre Nosotras, providing actionable guidance to enhance MHPSS intervention completion by refugee, migrant, and host communities. Most importantly, our findings underscore the importance of contextually tailorizing MHPSS interventions during the early design stages to further adapt for the heterogeneous settings in which these interventions are delivered.

### Limitations

While our analysis successfully identified several sources of variation in the reasons for session nonattendance and intervention retention across and within study sites, it is important to acknowledge certain limitations. Our analysis was unable to capture other details of the variations in the implementation processes across individual communities, which may have led to differences in intervention delivery across sites and influenced completion status. These variations could stem from differences in recruitment procedures and the staff responsible for implementing the interventions within each site. Additionally, the COVID-19 pandemic affected each site differently, leading to varying implementation timelines and procedures, including the occasional transition to online sessions. Although logistical issues ranked lowest among reasons for missing any session across all sites, specific data on sessions conducted remotely and their impact on session completion are not readily available, limiting our ability to fully assess their potential influence. 

## 5. Conclusions

Our analysis of intervention completion provides valuable insights into enhancing retention within MHPSS interventions, which ultimately leads to their successful implementation. The observed differences in completion rates and reasons for nonattendance across sites suggest that tailoring MHPSS interventions will require adaptations that further consider the duration of access to the target population, and explore different modalities for intervention delivery and the continued engagement of participants. Additionally, greater attention must be paid to engaging with younger, newly arrived women. Finally, to facilitate attendance at sessions, strategies such as consulting scheduling preferences, offer in site childcare services, and integrating MHPSS interventions with other support programs must be considered. Future research should focus on understanding what is behind the participants being “unreachable” and exploring the role of cellular devices as a reliable or unreliable tool to maintain communication with study participants and possibly deliver MHPSS interventions.

## Figures and Tables

**Table 1 ijerph-21-00397-t001:** Characteristics of Entre Nosotras Study Population by Completion Status.

Characteristic	Overall *n* = 225 (100.0%)	Completed Intervention (4–5 Sessions) *n* = 112 (49.8%)	Partial Attendance (1–3 Sessions) *n* = 63 (28.0%)	Never Started (0 Sessions) *n* = 50 (22.2%)	*p*-Value
Age Mean (SD)	36.0 (11.7)	38.9 (12.7)	32.9 (9.8)	33.6 (9.9)	0.001
Site *n* (%)					0.030
Panama	82 (36.4%)	49 (43.8%)	23 (36.5%)	10 (20.0%)	
Guayaquil	72 (32.0%)	28 (25.0%)	24 (38.1%)	20 (40.0%)	
Tulcán	71 (31.6%)	35 (31.2%)	16 (25.4%)	20 (40.0%)	
Nationality *n* (%)					0.2
Venezuelan	147 (65.9%)	67 (60.4%)	44 (69.8%)	36 (73.5%)	
Colombian	33 (14.8%)	17 (15.3%)	11 (17.5%)	5 (10.2%)	
Ecuadoran	28 (12.6%)	20 (18.0%)	3 (4.8%)	5 (10.2%)	
Other	15 (6.7%)	7 (6.3%)	5 (7.9%)	3 (6.1%)	
Education *n* (%)					0.3
High school	116 (52.0%)	58 (52.3%)	27 (42.9%)	31 (63.3%)	
University degree	55 (24.7%)	25 (22.5%)	20 (31.7%)	10 (20.4%)	
Elementary school or less	38 (17.0%)	18 (16.2%)	13 (20.6%)	7 (14.3%)	
Other	14 (6.3%)	10 (9.0%)	3 (4.8%)	1 (2.0%)	
Employment *n* (%)					0.3
Unemployed	120 (53.8%)	67 (60.4%)	32 (50.8%)	21 (42.9%)	
Informal worker	68 (30.5%)	28 (25.2%)	20 (31.7%)	20 (40.8%)	
Formal worker	35 (15.7%)	16 (14.4%)	11 (17.5%)	8 (16.3%)	
Migration Reason *n* (%)					0.005
Economic troubles	85 (42.3%)	37 (37.0%)	26 (44.8%)	22 (51.2%)	
Family reasons	61 (30.3%)	37 (37.0%)	11 (19.0%)	13 (30.2%)	
Violence or conflict	28 (13.9%)	12 (12.0%)	14 (24.1%)	2 (4.7%)	
For work	19 (9.5%)	6 (6.0%)	7 (12.1%)	6 (14.0%)	
Others	8 (4.0%)	8 (8.0%)	0 (0.0%)	0 (0.0%)	
Unknown	24	12	5	7	
Time in community *n* (%)					0.052
Less than 1 year	53 (23.6%)	22 (19.6%)	15 (23.8%)	16 (32.0%)	
1–3 years	90 (40.0%)	39 (34.8%)	31 (49.2%)	20 (40.0%)	
>3 years	82 (36.4%)	51 (45.5%)	17 (27.0%)	14 (28.0%)	

Age was compared across intervention completion groups using One-way ANOVA. For categorical variables such as study site, participant’s nationality, education, employment, reasons for migration, and time in the community, Pearson’s Chi-squared test or Fisher’s Exact Test for Count Data with simulated *p*-values (based on 2000 replicates) were employed. There is missing information for age in one observation and for nationality, education, and employment, in two observations.

**Table 2 ijerph-21-00397-t002:** Description of Reasons for Missing any Entre Nosotras Session.

Reason	Description
Unreachable	Staff members could not communicate with participant to assess cause for non-attendance.
Work or school	Participant missed the intervention session because they had to go to work or school.
Other personal causes	Participant mentioned having a personal inconvenience. This could include running errands, not having enough economic resources to afford transportation, and traveling.
Medical incapacity	Participant reported not feeling well, being sick, having to attend a medical appointment, or being hospitalized.
Family responsibilities	Participant did not attend the interventions session because they had to take care of family members.
Logistical issues	Participant could not attend because they faced barriers for getting to session due to external factors. These include rainy weather, or difficulties finding their address. Occasionally, some communities held sessions online, for which issues related to connectivity and technological devices are also included in this category.

**Table 3 ijerph-21-00397-t003:** Reasons for missing any session by study site.

	Overall Missed Sessions *n* = 484	Guayaquil *n* = 185	Panamá *n* = 138	Tulcán *n* = 161
Reason *n* (%)				
Unreachable	125 (27.8%)	46 (29.1%)	17 (12.6%)	62 (39.7%)
Work or school	110 (24.5%)	50 (31.6%)	40 (29.6%)	20 (12.8%)
Other personal causes	84 (18.7%)	21 (13.3%)	39 (28.9%)	24 (15.4%)
Medical incapacity	64 (14.3%)	15 (9.5%)	19 (14.1%)	30 (19.2%)
Logistical issues	18 (4.0%)	6 (3.8%)	5 (3.7%)	7 (4.5%)
Unknown	35	27	3	5

**Table 4 ijerph-21-00397-t004:** Main reasons for intervention non-completion for participants who partially attended and never started.

	Overall Intervention Non-Completion *n* = 113	Partial Attendance *n* = 63	Never Started *n* = 50
Main Reason *n* (%)			
Combination of reasons	26 (23.0%)	22 (34.9%)	4 (8.0%)
Unreachable	26 (23.0%)	5 (7.9%)	21 (42.0%)
Work or school	26 (23.0%)	13 (20.6%)	13 (26.0%)
Other personal causes	14 (12.4%)	7 (11.1%)	7 (14.0%)
Medical incapacity	12 (10.6%)	9 (14.3%)	3 (6.0%)
Family responsibilities	6 (5.3%)	4 (6.3%)	2 (4.0%)
Logistical issues	3 (2.7%)	3 (4.8%)	0 (0%)

## Data Availability

Data will be made publicly available through a data repository managed by the MHPSS knowledge hub (https://mhpssknowledgehub.sph.cuny.edu/ accessed on 27 July 2022). At the time of submission, the data repository was still in development. Prior to its establishment, data and statistical programs may be made available upon reasonable request to the primary author.

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
