# Peer review of "Improving Retention in Mental Health and Psychosocial Support Interventions: An Analysis of Completion Rates across a Multi-Site Trial with Refugee, Migrant, and Host Communities in Latin America"

_ijerph, 2024, doi:10.3390/ijerph21040397_

Round 1
Reviewer 1 Report
Comments and Suggestions for Authors
Major points
The manuscript describes an important study on monitoring participation in an intervention program for displaced women but the description of the participants is not clear for 2 reasons.
1. The title uses the word „displaced” and „host” populations; in the Introduction, „forced migration” is mentioned; HIAS is a „refugee protection organization”, in relation to the Entre Nosotras study, the term „migrants” is used. Description of the target group is missing from „Objectives”. „Refugee” and „migrant„ refer to different population groups with not necessarily overlapping problems so these must be distinguished. „Forced migration” is an unfamiliar term that the authors must also clarify.
2. The authors write about Venezuelan migrants being the largest displaced population in the first paragraph of Introduction, then describe the Entre Nosotras program that aimed „displaced and host community women in Ecuador and Panama” (L 85-86), but it is not clear who the „host” (citizen in the country where she resides?) women are. Migrant or host status is not listed among the eligibility criteria described in „Participants and procedures”. So who were included in this study, only migrants or „host” women as well?
What was the total number of participants in Entre Nosotras? How were participants selected (especially those who did not attend any sessions) for this study and when was their psychogical distress assessed? Given that the level of stress was an inclusion criterion, this information should be uncovered.
Why only „up to moderate” psychological distress (L124) rendered participants eligible, why those with severe distress were excluded?
The authors should give more details about the interviews by which they gathered data from those who missed any sessions even more so because as it seems from Table 3, the largest group of missing participants was unreachable.
There are ” 1 ” notes below the Completion categories in Table 1 and these are explained as ”Mean (SD)” below which the percent distribution of participants by completion status is given. This is not comprehensible, the mean of what is referred to at the top of the columns?
The major question related to Results is why the authors were more interested in the characterictics of participants by SITES of the interventions instead of by COMPLETION STATUS if retention or as they stated in the Objective, „reasons for non-attendance” was their primary question?
Table 1 does not allow any simple inference for retention because the authors gave the percent distribution of participant characteristics separately for each category of completion status (completed intervention in Panama, om Guayaquil, in Tulcan, etc.) rather than by categories of participant characteristics (Completed, partial attendance, never started in Panama, etc). The reader therefore is not fully convinced that their statistical analysis in this table is correct because:
1) The index (2) of the last column (p-value) lists a number of statistical procedures and it is not clear which test was used for what indicator (eg. age (missing measure and unit) was hopefully not tested by chi2 test; site distribution was surely not tested by ANOVA?)
2) Even if the reader supposes that p values of the categorical characteristics derive from chi2 or Fisher exact tests, no meaningful conclusion can be drawn as to the differences in nationality, education, etc. among those who completed, partially completed or never started the intervention. Why did the authors cross-tabulate the data this way instead of investigating participant characteristics by categories of attendance?
3) What does the „other” category of education mean? Why education is not defined as an ordinal (not nominal) variable?
4) Why is the „unknown” category of migration reason not given in percentage?
5) The lack of clear definition of terms can be deduced from Table 1: why violence or conflict as a reason for migration does not make someone a refugee?
The authors tabulated the reasons for missing by sessions (Table 3) and by reason for missing (Table 4). Interpretation of Table 3 is limited because the authors did not specify the grand total number of sessions from which 484 were missed. It is not fully comprehensible for the reader whether the interviews asked separate questions about the reason for missing every single session separately (shown in Table 3) or missing sessions in general (Table 4) or both.
As it can be seen from the data, 112/225=50% of the women did not fully complete the intervention, and 50/225=22% never started it. However, the authors did not analyze how the demographic characteristics of participants shown in Table 1 are correlated with reasons for missing sessions shown in Table 4. Therefore, taken together with the curious method of cross-tabulation described for Table 1, their conclusions – with the exception of „non-completion women” being younger – are not supported by their data, and the authors’ conclusions do not include important demographic characteristics such as education, employment status and reasons for migration as potential determinants of non-participation.
The reviewer found it interesting that the authors did not collect data on the characteristics of those who delivered the interventions and circumstances of the interventions. Moreover, from the way they structured their data in the Tables, they seemed to hypothethize that interventions were uniform in all sites, a highly unlikely scenario.
Minor remarks:
What is HIAS?
L63 preform differently – not perform?
L86 intervention principals – not intervention principles?
Table 3: The percentage of missed sessions for unknown reasons is not given. The superscripts „1” make no sense in the table heading of because it is given in the cells (except the last row), and is irrelevant for the heading where absolute numbers are shown after „N”.
Comments on the Quality of English LanguageAcceptable.
Reviewer 2 Report
Comments and Suggestions for Authors
This is a well-written important article, the results of which can be applied to other similar programs world-wide.
I have just a few comments.
1. Line 63 – typo – preform should be perform.
2. Line 124 - Describe the Kessler scale – perhaps a sentence.
3. Table 1 – Identify age as mean(SD)
4. Table 1 – Sort time in community from least to greatest (LT 1 year, 1-3 years, GT 3 years).
5. Results – for identifying p-values within the text, there is no need to identify the test with the p-value (e.g. One-way ANOVA p=0.001).
6. Table 3 – Instead of including n(%) as a note, include it in the headings for the table.
7. Discussion/conclusion – Provide narrative of how the information gained from this study will assist other similar programs, beyond the Entra Nosotras.
Reviewer 3 Report
Comments and Suggestions for Authors
Thank you very much for giving me the opportunity to review the manuscript entitled “Improving Retention in Mental Health and Psychosocial Support Interventions: An Analysis of Completion Rates Across a Multi-Site Trial with Displaced and Host Populations in Latin America”. I have read it with great interest. This secondary analysis examines differences of the participant characteristics, study setting, and reasons for intervention noncompletion by their completion status. It was found that participants who were older, had migrated for family reasons, had spent more time in the study community, and were living in Panamá (vs. Ecuador) were more likely to complete the intervention. This study deals with a very important but currently under-studied topic: retention of participants in mental health and psychosocial support interventions in humanitarian settings. I highly appreciate the research team’s effort to conduct this study. The submitted manuscript is well written. I have a few minor suggestions to improve readability.
1) Line 103. This paragraph explains Guayaquil, Tulcán, and Panamá City/Panamá West. It seems better to add more information about employment situation such as unemployment rate. This is because working or studying is one of main reasons for non-participation.
2) Line 137, “The responses to this open- ended question were recoded into categories in the reason for missing variable”. It seems better to explain how this open- ended question were recoded into categories. How many research staffs did participate in this categorization?
3) Line 211, “Table 3. Reasons for Missing any Session by Study Site”. It seems better to put “1n (%)” below the line.
4) Line 229, “Frequently, reporting on retention of participants in MHPSS intervention is omitted”. It might be better to add “in humanitarian settings” after “MHPSS interventions”. It is custom practice to report retention rate in RCT in other mental health area.
5) I can not understand the numbers in the first line of Table 3. For example, it is difficult to read “Overall missed sessions, N =1 484”. I appreciate if you would make necessary changes.
Author Response
Please see the attachment,
